Unravelling potential virulence factor candidates in Xanthomonas citri. subsp. citri by secretome analysis

Ferreira Rafael M. 1 marini64@gmail.com
Moreira Leandro M. 2
Ferro Jesus A. 1
Soares Marcia R.R. 3
Laia Marcelo L. 4
Varani Alessandro M. 1 amvarani@fcav.unesp.br
de Oliveira Julio C.F. 5
Ferro Maria Ines T. 1
1 Departamento de Tecnologia, Universidade Estadual Paulista “Júlio de Mesquita Filho” , Jaboticabal, São Paulo , Brazil
2 Departamento de Ciências Biológicas—Núcleo de Pesquisas em Ciências Biológicas-NUPEB, Universidade Federal de Ouro Preto , Ouro Preto, Minas Gerais , Brazil
3 Departamento de Bioquímica, Universidade Federal do Rio de Janeiro, Instituto de Química , Rio de Janeiro, Rio de Janeiro , Brazil
4 Departamento de Engenharia Florestal, Universidade Federal dos Vales do Jequitinhonha e Mucuri , Diamantina, Minas Gerais , Brazil
5 Departamento de Ciências Biológicas, Universidade Federal de São Paulo , Diadema, São Paulo , Brazil
Lazo Gerard
Electronic publication date: 2016 Feb 23
Publication date: 2016
Volume: 4
Electronic Location ID: e1734
Received 2015 Nov 19; Accepted 2016 Feb 2
Copyright: ©2016 Ferreira et al.
Copyright year: 2016
Copyright holder: Ferreira et al.
License: This is an open access article distributed under the terms of the Creative Commons Attribution License, which permits unrestricted use, distribution, reproduction and adaptation in any medium and for any purpose provided that it is properly attributed. For attribution, the original author(s), title, publication source (PeerJ) and either DOI or URL of the article must be cited.
License URL: https://creativecommons.org/licenses/by/4.0/

Keywords: Plant–pathogen interaction, Medium inducing pathogenicity, Type III secretion system, Virulence, Asymptomatic mutant, Type II secretion system, Secretomics

Funding: CAPES—BIGA (Coordenação de Aperfeiçoamento de Pessoal de Nível Superior) CAPES post-doctoral fellowship This work was supported by grants from CAPES—BIGA (Coordenação de Aperfeiçoamento de Pessoal de Nível Superior) project. RMF was funded by a CAPES post-doctoral fellowship. The funders had no role in study design, data collection and analysis, decision to publish, or preparation of the manuscript.

==============================
Citrus canker is a major disease affecting citrus production in Brazil. It’s mainly caused by Xanthomonas citri subsp. citri strain 306 pathotype A (Xac). We analysed the differential expression of proteins secreted by wild type Xac and an asymptomatic mutant for hrpB4 (ΔhrpB4) grown in Nutrient Broth (NB) and a medium mimicking growth conditions in the plant (XAM1). This allowed the identification of 55 secreted proteins, of which 37 were secreted by both strains when cultured in XAM1. In this secreted protein repertoire, the following stand out: Virk, Polyphosphate-selective porin, Cellulase, Endoglucanase, Histone-like protein, Ribosomal proteins, five hypothetical proteins expressed only in the wild type strain, Lytic murein transglycosylase, Lipoprotein, Leucyl-tRNA synthetase, Co-chaperonin, Toluene tolerance, C-type cytochrome biogenesis membrane protein, Aminopeptidase and two hypothetical proteins expressed only in the ΔhrpB4 mutant. Furthermore, Peptidoglycan-associated outer membrane protein, Regulator of pathogenicity factor, Outer membrane proteins, Endopolygalacturonase, Chorismate mutase, Peptidyl-prolyl cis-trans isomerase and seven hypothetical proteins were detected in both strains, suggesting that there was no relationship with the secretion mediated by the type III secretory system, which is not functional in the mutant strain. Also worth mentioning is the Elongation factor Tu (EF-Tu), expressed only the wild type strain, and Type IV pilus assembly protein, Flagellin (FliC) and Flagellar hook-associated protein, identified in the wild-type strain secretome when grown only in NB. Noteworthy, that FliC, EF-Tu are classically characterized as PAMPs (Pathogen-associated molecular patterns), responsible for a PAMP-triggered immunity response. Therefore, our results highlight proteins potentially involved with the virulence. Overall, we conclude that the use of secretome data is a valuable approach that may bring more knowledge of the biology of this important plant pathogen, which ultimately can lead to the establishment of new strategies to combat citrus canker.

Introduction

Proteins secreted by bacteria are known to have key functions, such as the provision of nutrients, cell–cell communication, detoxification, inhibition of potential competitors, etc. (Tseng, Tyler & Setubal, 2009). The extracellular proteins of pathogenic bacteria also play a critical role in their pathogenicity and adaptation to a compatible host (Kamoun et al., 1993; Qian et al., 2013).

Six types of secretory systems (SS), which have been previously described, are critical for exporting proteins to the external environment or for directly contacting the host (Tseng, Tyler & Setubal, 2009). SS can be classified into two groups: (I) one-step secretion systems, wherein secreted proteins are exported across the inner and outer membranes in a single step, include the type I (T1SS), type III (T3SS), Type IV (T4SS), and type VI (T6SS) secretion systems; or (II) two-step secretion systems, wherein proteins are first exported into the periplasmic space via the general secretion (Sec) or two-arginine (Tat) pathways, and then translocated across the outer membrane via the type II (T2SS), type V (T5SS), or less commonly, the T1 or T4SS systems (Saier, 2006).

Secreted proteins (secretomes) from different organisms cultivated in different physiological conditions have been analysed, which often validates biological information derived from genomic sequencing projects. Previous studies have focused on the secretomes of specialized secretion apparatuses, which include the Tat (Pickering, Yudistira & Oresnik, 2012), T3SS (Deng et al., 2010) and T2SS secretomes (Sikora et al., 2011). In other studies, the overall goal was to characterize the secretome irrespective of the secretion apparatus used (Zijnge, Kieselbach & Oscarsson, 2012). More recently, comparative proteomics, which aims to compare the secretomes of different bacterial strains cultivated in the same physiological conditions (Xu et al., 2013) or the secretomes of the same strain cultivated in different physiological conditions or hosts (Villeth et al., 2009), has gained traction.

The secretomes of plant pathogenic bacteria have also been studied (Kazemi-Pour, Condemine & Hugouvieux-Cotte-Pattat, 2004; Nissinen et al., 2007; Schumacher et al., 2014), including bacteria of the genus Xanthomonas (Evans et al., 2007; Wang et al., 2013). Xanthomonas citri subsp. citri (Xac), strain 306 pathotype A, which causes canker A, affects different citrus species. The study of its general proteome have contributed to a greater understanding of its pathogenicity and adaptation (Yamazaki, Hirata & Tsuyumu, 2008; Soares et al., 2010; Facincani et al., 2014; Moreira et al., 2015). Since Xac is a quarantine microorganism that is the cause of numerous losses to the citrus industry, characterization of its genome, proteome, and secretome are important for understanding the mechanisms of its interaction with compatible hosts.

In Xac, as well as in other bacteria of the same genus, the T3SS has been the best studied SS. Through this apparatus, specific effector proteins of virulence are secreted (Büttner & He, 2009), which mediate the typical symptoms of the disease in their respective susceptible plant hosts. Previous studies have shown that mutations in genes that are part of or regulate this apparatus completely prevent or substantially reduce the classical phenotypes of the disease (Büttner et al., 2002). Random mutations in Xac were generated previously by our group using a EZ-Tn5 <Kan>transposition cassette, which were used to identify several genes that might be associated with its virulence (Laia et al., 2009). Of these, one mutation in particular resulted in non-pathogenicity; the mutation was in hrpB4 (ΔhrpB4-XAC0410), located within the hrp (Hypersensitive Response and Pathogenicity) gene cluster and coding for the T3SS apparatus (Fig. 1A). HrpB4 is the inner membrane protein that associates with T3SS, thus a structural protein that plays a role on T3SS apparatus assembly (Fig. 1B). Interestingly, in addition to being unable to induce virulence (Fig. 1C), ΔhrpB4 also fails to develop within the plant despite growing normally in an energy-rich culture medium. At the same time, because ΔhrpB4 was unable to grow in planta, it would not be possible to obtain the secreted proteins in infective conditions in concentrations capable of being analysed by proteomics.

Figure 1 (A) Cluster of hrp genes in the Xac genome, highlighting the mutation of the hrpB4 gene induced by insertion of the transposition cassette EZ-Tn5 <Kan >. The insertion was made at position 147 in the gene. (B) Structural composition of the corresponding apparatus to the type III secretion system encoded by hrp genes. The HrpB4 protein is highlighted in red. (C) Phenotype virulence alteration of the mutant ΔhrpB4 (complete absence of symptoms) compared to the wild-type strain (WT) after 12 days of infection in Citrus sinensis plants.

To characterize the secretome of Xac, we cultured the wild-type strain and the ΔhrpB4 mutant in two different culture media: nutrient broth (NB) and the defined medium, XAM1 (Wengelnik & Van den Ackerveken, 1996). The latter medium was chosen based on previous results showing that XAM1 medium simulates conditions experienced by Xac during plant infections (Facincani et al., 2014). The mutant strain, in which the T3SS is non-functional, was chosen to test the dependence of the secretome on the T3SS. In addition, since several authors have previously reported a relationship between xanthan gum production and the pathogenicity of the microorganism (Katzen et al., 1998), assays to determine the production of xanthan gum and the dry weight of cells were performed at each condition.

Materials and Methods

Growth conditions

Stocks of wild-type Xac and the ΔhrpB4 mutant strain were stored in phosphate-buffered saline at room temperature. Both were cultured in Nutrient Agar plates prior to the preparation of the pre-inoculum. All maintenance cultures were maintained at 28 °C, and kanamycin was added at 100 µg/mL for solid culture media and 50 µg/mL for liquid culture media. Wild-type Xac and ΔhrpB4 were cultured in a 15-mL liquid Nutrient Broth (NB) pre-inoculum, maintained at 200 rpm and 28 °C, until the cultures reached OD600nm = 1.0 (approximately 108 CFU/mL). Cultures were centrifuged at 3,100 × g for 15 min and the supernatant was discarded. The pellet was resuspended in water and added to the culture medium. Both wild-type Xac and ΔhrpB4 were grown in 500 mL NB culture medium (200 rpm, 28 °C) until the culture reached an OD600nm of at most 1.4 (approximately 109 CFU/mL). At OD600nm less than 1.4, growth is still logarithmic, which prevents the release of proteins into the culture medium due to cell lysis, thus enabling us to accurately profile the secretome (Alexander Watt et al., 2005). XAM1 media (7.5 mM (NH4)2SO4, 33 mM KH2 PO4, 60 mM K2 HPO4, 1.7 mM sodium citrate [C6 H5 Na3 O7⋅2H2O], 0.9 mM MgSO4, 9.9 mM fructose, 9.9 mM sucrose and 0.03% of Casamino acids in a final volume of 1 L) was used to induce virulence (Wengelnik & Van den Ackerveken, 1996). Each bacterium was grown in 2 L (4 × 500 mL) of this medium, to which 1 mg/mL of Bovine Serum Albumin (BSA) was added. Bacteria were grown at 28 °C with shaking at 200 rpm until the culture reached an OD600 of at most 0.57, wherein growth was still logarithmic.

Wild-type Xac or ΔhrpB4 were grown for 16 or 23 h in NB liquid culture media until it reached an OD600 of 1.0 or 1.4, respectively. Both strains were also grown in XAM1 liquid media for 24 h, which is equivalent in time to the early stages of infection in vivo. Wild-type bacteria reached an OD600 of 0.57, and the mutant achieved an OD600 of 0.24. Higher cell densities in XAM1 liquid media could not be achieved even with longer cultivation periods (>24 h). Thus, to achieve the highest density of bacteria without compromising their structural integrity, a cultivation time of 24 h was chosen, which corresponded with the initial stage of infection. The difference in growth between the two culture media is due to the fact that the first is an energy-rich medium and the second a defined medium.

Several replicates were used for each condition to obtain the total pool of proteins used for this study.

Centrifugation and filtration of samples

Cultures were centrifuged in two 60 min steps at 4 °C. In the first step, the samples were centrifuged at 3,100 × g, the bacterial pellet was discarded and the supernatant containing secreted proteins was transferred to clean tubes. In the second step, the (first) supernatant was centrifuged at 11,000 × g and again the supernatant was recovered. For strains cultivated in NB liquid medium, the supernatant was recovered and then filtered using 0.2-µm (Millipore™ ) nitrocellulose membranes shortly after the second centrifugation.

For bacteria cultured in XAM1 medium, the supernatant could not be filtered due to the elevated production of xanthan gum, which caused changes in the membrane filtration profile. All samples were frozen using liquid nitrogen and lyophilized to complete dryness. Each dried sample corresponding to 500 mL of the supernatant was resuspended in 50 mL of double-distilled autoclaved water (ddH2O). To precipitate xanthan gum produced by cultures grown in XAM1 medium, absolute ethyl alcohol (1:3) was added to the resuspended sample (in ddH2O), and the sample was centrifuged at 3,000 × g for 5 min at 30 °C. These samples were then filtered through 0.2 µm (Millipore™) nitrocellulose membranes and again lyophilized and resuspended in 50 mL of ddH2O.

Precipitation of proteins and SDS-PAGE electrophoresis

Proteins in the filtered supernatant were precipitated with trichloroacetic acid (TCA), with modifications (Hirose et al., 2000). Briefly, TCA was added to the resuspended sample to a final concentration of 10%, and each sample was vigorously mixed and stored at 4 °C for 16 h. Samples were centrifuged at 20,000 × g for 20 min at 4 °C. The supernatant was discarded and the precipitate washed three times with cold acetone to remove residual TCA. Precipitated proteins were then lyophilized to complete dryness and dissolved in 25 mM NH4HCO3 buffer, 500 mM urea, pH 8.0. All samples were quantified and normalized. The proteins were then hydrolysed in solution.

Trypsinization of samples and mass spectrometry

Proteins were reduced by adding 1 µg of DTT per 50 µg of protein and incubating for 1 h at 37 °C. Proteins were then alkylated by adding 5 µg of iodoacetamide per 50 µg of protein and incubating for 1 h at room temperature in the dark. Proteins were trypsinized by adding 20 µg of Promega Sequencing Grade Modified Trypsin (Madison, USA) (1:50) and incubating for 20 h at 37 °C. The hydrolysis reaction was terminated by adding 2 µL of formic acid. The samples were then loaded into a Waters® nanoACQUITY UPLC® capillary chromatography system (Waters, Milford, MA). The digested proteins were desalinized using a Waters Opti-Pak C18 trap column. The volume of the injected sample was 10 µL, and liquid chromatography was performed on a reverse-phase C18 column Ease 150 mm × 2.1 mm (Waters, Milford, MA). Proteins were eluted with a flow rate of 0.3 µL/min using a linear gradient ranging from 5% to 50% acetonitrile containing 0.1% formic acid.

The capillary liquid chromatography system (nLC) was coupled to a mass spectrometer with an electrospray ionization (ESI) source and quadrupole/time-of-flight in series analysers (Q-TOF Micro; Waters, Milford, MA) (UPLC-MS/MS), which permitted the direct analysis of eluted peptides in the acetonitrile gradient in Q-TOF. For ESI, a voltage of 3,000 V and a temperature of 80 °C was applied in the capillary. A MassLynx data system (Version 4.1, Waters) was used to control the instrument and acquire data, and the experiments were performed by scanning mass/charge (m/z) ratios between 200 and 2,000 using a first scan time of 1 s, which was applied during the entire chromatographic process. The averages of the mass spectra corresponding to each sign of the total ion current chromatogram (TIC) were calculated, allowing an accurate determination of the molecular mass. Exact mass values were obtained using a LockSpray™ source (Waters, Milford, MA). This reference mass was used to correct the mass of the analyte (sample) throughout the study. Phosphoric acid was used as the reference in this study, which has an m/z ratio of 588.8692.

The acquisition of data dependent on MS/MS were performed on precursors with charge states 2 or 3 over a mass range of m/z 50–2,000 and at an interval below 2 m/z. At most, 3 ions were selected for MS/MS analysis of a single MS. The masses of Na+ and K+ were automatically excluded. The collision induced dissociation (CID) based MS/MS spectra were obtained by using argon as the collision gas at a pressure of 13 PSI and a collision voltage that ranged from 18 to 45 V, depending on the mass of the precursor. The scanning ratio was 1 s.

Processing of the generated spectra and signal peptide prediction

All data were processed with ProteinLynx Global Server (version 2.0, Waters), which automatically corrected values of m/z spectrum of MS and MS/MS according to the mass of the Lockspray reference ion. Proteins were identified using the Xac database from NCBI (BioProject PRJNA297) and UniProt databases, and the MASCOT program (Version 2.2.1; Brazilian Synchrotron Light Laboratory). The values of monoisotopic masses of the MS/MS spectra (MS/MS Ion Search) were used, which considered cysteine carbamidomethylation as a fixed modification and oxidation of methionine as a variable modification. In the hydrolysis by trypsin, the possible loss of one cleavage site was considered and the tolerance of masses of peptides and fragments was set to ± 0.05 Da.

The amino acid sequences of identified proteins were submitted to analysis by PrediSi (Prediction of Signal peptides, www.predisi.de) (Hiller et al., 2004). Identified proteins that lacked signal peptides according to PrediSi analysis were submitted to SecretomeP 2.0 for further analysis (Bendtsen et al., 2005). Comparisons between proteins from different organisms were performed using the bioinformatics tool BLASTP.

Identification of candidate PIP-boxes in the regulatory regions of genes

Several genes regulated by HrpX possess a consensus nucleotide sequence, TTCGC-N15-TTCGC, which has been termed the plant-inducible promoter box (PIP-box) (Da Silva et al., 2002). The presence of a PIP-box or PIP-box-like sequence suggests that the candidate gene is regulated by HrpX and encodes an effector protein of the T3SS pathway. PIP-box analysis was performed following methods described by Da Silva (Da Silva et al., 2002).

Candidate PIP-box sequences and their genomic locations were identified by searching the complete Xac genome for sequences that matched the following PIP-box sequence patterns: 1. Forward orientation (uncomplemented)—TTCGN-N15-TTCGN or TTCGN-N16-TTCGN or 2. Reverse orientation (complemented)—NCGAA-N15-NCGAA or NCGAA-N16-NCGAA. Each candidate PIP-box sequence was further analysed to determine whether it was part of a promoter, which should be between 10 and 1,000 nucleotide bases long and precede the start codon of a gene.

Xanthan gum production

For xanthan gum production, three (triplicate) 250-mL Erlenmeyer flasks containing 100 mL of Gum medium (25 g/L glucose, 3 g/L yeast extract, 2 g/L K2HPO4, 0,1 g/L MgSO4.7H2O, pH 7.0 with 4 M HCl) were inoculated with 2.5 mL of each bacterial strain grown in NB liquid medium (OD600 of 0.3) and incubated at 29 °C for 96 h in a rotary shaker at 178 rpm (Shu & Yang, 1990).

After 96 h, cultures were centrifuged at 9,666 × g for 40 min. The bacterial pellets were stored at −20 °C and the supernatants were transferred to 500-mL beakers. Xanthan gum was recovered from the supernatants by isopropanol precipitation. KCl (4 g) was added to each beaker, followed by stirring at room temperature for 15 min. Two volumes of cold isopropyl alcohol were added and the xanthan gum from each beaker was removed and placed in plastic containers, which were previously weighed, with the aid of a glass rod and a sieve. After 48 h at 37 °C, the containers, plus sample, were weighed again and the amount of xanthan gum calculated. The mass of the bacterial pellet from each culture was also measured. For this, each pellet was resuspended with 1 mL MilliQ autoclaved water and transferred to a pre-weighed beaker and weighed again after 24 h at 70 °C. Three replicates were used for each condition and Statistical Analysis was performed (Unpaired Student t test).

Results and Discussion

Fifty-five unique proteins were identified in total on the secretomes of wild-type Xac and the mutant ΔhrpB cultured in different growth media. Of these, the majority (37 = 67.3%) were secreted by both strains only when cultured in XAM1 medium, whereas 12 (21.8%) were secreted only when cultured in NB (Fig. 2A). Considering that the XAM1 medium simulates conditions experienced by Xac during plant infections or when Xac is in contact with the plant meristem, the former set of proteins are considered the repertoire of proteins required for pathogenicity and adaptation during infections. Regardless of the growth media, 19 proteins (34.5%) were secreted solely by the wild-type strain, whereas 14 (25.5%) were secreted solely by the mutant strain (Fig. 2A). These differences may be directly attributed to the non-functionality of the T3SS system in the mutant ΔhrpB4. It is also possible that the functional loss of the T3SS apparatus, which may be important for maintaining cellular homeostasis and for adaptation in plant tissues, has pleiotropic effects on the expression/secretion of certain proteins.

Figure 2 Comparative analysis of Xac secretome.

(A) Comparison highlighting 55 total detected proteins, correlating culture mediums (XAM1 × NB—A1) and strains (Xac and Δ hrpB4—A2) analysed. The two circles represent the total proteins detected. (B) Categorization of annotated protein functions, adapted from Da Silva et al. (2002): I, Intermediary metabolism; II, Biosynthesis of small molecules; III, Metabolism of macromolecules; IV, Cell structure; V, Cellular processes; VI, Mobile genetic elements, VII, Pathogenicity; virulence, and adaptation, VIII, Hypothetical/Conserved hypothetical genes; XI, Without a defined function. (C) Venn Diagram highlighting each of the proteins detected under the four different conditions tested. Proteins annotated as hypothetical are highlighted in red. The same Venn diagram is shown on a smaller scale but with only the numbers of the proteins detected in each condition.

The Venn diagram shown in Fig. 2C underscores the differences and overlap in proteins secreted under the different conditions tested. Analysis of proteins according to their MASCOT annotated functions revealed that seven were associated with adaptation and virulence (category VII), and 12 were hypothetical proteins (category VIII) (Fig. 2B). The following sections describe in details these main findings.

Proteins secreted by both strains grown in XAM1 medium

Table 1 lists the 13 proteins secreted in common by both strains. Of these, four are hypothetical proteins (XAC2562, XAC0677, XAC0223 and XAC0232), three of which contain a signal peptide sequence (Table 1). Among the other nine proteins with annotated functions (Pcp, Cysteine protease, RpfN, Slp, OmpP6, Peh-1, PheA and Prolyl-peptidyl cis-trans isomerase—PPlase), six have a signal peptide sequence, and only the gene encoding for Peh-1 is downstream of a PIP-box sequence (Table 1). All of these membrane proteins were previously identified in the formation of outer membrane vesicles (OMVs) when X. campestris pv. campestris was grown in XVM2 medium (Sidhu et al., 2008), which is similar to XAM1 in that it also simulates conditions experienced by the microorganism during infections. In general, proteins secreted by both wild-type Xac and ΔhrpB4 in these conditions likely represent a core set of proteins required to survive and adapt to the plant interior. Importantly, the loss of T3SS function in the hrpB4 mutant did not affect the secretion of these proteins; therefore, it is likely that these proteins are secreted by the other secretory systems, such as Sec, Tat, T2SS, and T5SS. The presence of signal peptide sequences in at least 69% of these proteins reinforces their targeted secretion.

Table 1 Proteins expressed/detected in wild type and ΔhrpB4 strains only in infectious conditions.

					Strains (culture media)			
Gene ID	Gene name	Product	Cat.	NCBI ID	ΔhrpB4 (NB)	Xac (NB)	ΔhrpB4 (XAM1)	Xac (XAM1)	SP	PIP	
XAC0223		Hypothetical protein	VIII	1154294	−	−	+	+	Y	N	
XAC1466	pcp	Peptidoglycan-associated outer membrane lipoprotein	IV	1155537	−	−	+	+	Y	N	
XAC2853		Cysteine protease	III	1156924	−	−	+	+	Y	N	
XAC2504	rpfN	Regulator of pathogenicity factors	VII	1156575	−	−	+	+	N*	N	
XAC1113	slp	Outer membrane protein Slp	IV	1155184	−	−	+	+	N*	N	
XAC2562		Hypothetical protein	VIII	1156633	−	−	+	+	Y	N	
XAC0677		Hypothetical protein	VIII	1154748	−	−	+	+	Y	N	
XAC3141	ompP6	Outer membrane protein P6	IV	1157212	−	−	+	+	N*	N	
XAC0552		Proteinase	III	1154623	−	−	+	+	Y	N	
XAC0661	peh-1	Endopolygalacturonase	VII	1154732	−	−	+	+	Y	Y	
XAC3647	pheA	Chorismate mutase	II	1157718	−	−	+	+	Y	N	
XAC0232		Hypothetical protein	VIII	1154303	−	−	+	+	N*	N	
XAC1585		peptidyl-prolyl cis-trans isomerase	III	1155656	−	−	+	+	Y	N	
Notes.

Gene_ID and Cat (primary category) according to Da Silva et al. (2002)

Product according to Kegg (Ogata et al., 1999)

SP Signal peptide

Y Yes

N No

N* Secreted by non-classical pathways

Proteins with annotated functions described bellow appear to have a fundamental role in either the virulence of the genus Xanthomonas or their adaptation to the plant. The Pcp protein, encoded by the gene XAC1466, is a peptidoglycan-associated outer membrane lipoprotein homologous to the SlyB protein in Xanthomonas campestris. This protein is directly regulated by a two-component system in most bacteria (PhoPQ), which is present in many bacterial pathogens of both animals and vegetables (Perez et al., 2009). This two-component system, which is encoded by the genes XAC4023 and XAC4022, is also expressed in Xac and has been shown to be active during the early stages of infection (Moreira et al., 2015). Proteins homologous to Pcp are also essential for maintaining the integrity of the cell envelope in Pseudomonas putida (Rodríguez-Herva, Ramos-González & Ramos, 1996), and more recently, were associated with OMV formation in P. aeruginosa (Wessel et al., 2013).

The Peh-1 protein, encoded by the gene XAC0661, is an endopolygalacturonase that facilitates plant cell wall degradation (Liu, Chatterjee & Chatterjee, 1994). Breakdown of the plant cell wall also releases sources of carbon to nourish the pathogen during infections. This protein is secreted by T2SS, and has been reported to be regulated by HrpG/X (Wang, Rong & He, 2008), which is consistent with the presence of the plant-inducible promoter (PIP-box) sequence located upstream of the gene. HrpG and HrpX are dual regulatory elements that regulate the expression of genes involved in the formation and secretion of T3SS proteins (Guo et al., 2011). In transcriptome studies, the hrpX gene was previously shown to be expressed in Xac when grown in XAM1 medium, and remains active even in the ΔhrpB4 strain (Laia et al., 2009), which is consistent with its secretion by both strains in our study. Mutants of Xanthomonas campestris in which the gene encoding the PghAxc protein (homologous to Peh-1 in Xac) was inactivated showed attenuated virulence when sprayed onto Arabidopsis hosts, demonstrating the importance of this protein in the early stages of infection (Wang, Rong & He, 2008).

A protein with cysteine protease functions, encoded by the XAC2853 gene, was also secreted by both strains grown in XAM1 medium. This protein is homologous to CysP2 in X. oryzae pv. oryzae, has a defective PIP-box sequence upstream of the gene in the Xac genome, is secreted by T2SS, and is regulated by HrpG/X (Furutani et al., 2004; Yamazaki, Hirata & Tsuyumu, 2008). Proteins with this function are considered important virulence factors because they hydrolyse peptides in the host cell. Inoculation of a citrus host in vivo with a mutant deficient in a cysteine protease, encoded by the gene XAC2853, resulted in less virulence (Soares-Costa et al., 2012). In studies involving Pseudomonas, the virulence of the pathogen was directly correlated with the secretion of the AvrRpt2 protein, which has cysteine protease functions, by T3SS (Axtell et al., 2003; Cui et al., 2013).

RpfN, which is characterized as a virulence factor regulator and encoded by the gene XAC2504, has an OprB domain (carbohydrate-selective porin). The OprB domain is a specific porin channel for glucose transport, but can also mediate the transport of other monosaccharides to the inner membrane (Wylie & Worobec, 1995). Pseudomonas aeruginosa mutants defective for this protein demonstrated a reduced ability to utilize various types of monosaccharides, indicating its importance in the import of different sugars into the cell (Nikaido, 2003). According to SecretomeP 2.0 analysis, this protein is not predicted to be secreted by the Sec pathway but rather by a non-classical route. The role of this protein in the pathogenicity of Xac is not well understood, but according to Dow and colleagues (2000), endopolygalacturonase levels were increased in Xanthomonas campestris mutants, in which this gene was inactivated. Moreover, Moreira and colleagues suggested that the absence of the rpfN gene and mutations in the PTS system associated with the internalization of fructose could explain the fastidious growth of X. fuscans B strain, which causes canker B in citrus plants (Moreira et al., 2010).

Slp, which is encoded by the gene XAC1113, is an outer membrane lipoprotein. Studies of homologues of this protein in the Xanthomonas genus are limited. However, in studies of the transcriptome of Xylella fastidiosa, which causes CVC (Citrus Variegated Chlorosis) in citrus plants, a homologue of the Xac slp gene was robustly expressed in growth conditions that induced the formation of biofilm (De Souza et al., 2004). Biofilms are fundamental to the induction of virulence and are produced on a large scale in bacteria of the genus Xanthomonas when grown in conditions that induce virulence (Dow et al., 2003; Rigano et al., 2007). Thus, the methodological difficulties of protein extraction from strains grown in XAM1 medium could be explained in part by the secretion of Slp proteins. In E. coli, changes to the composition of lipoproteins in the outer membrane directly affect surface adhesion, reinforcing the importance of these proteins to biofilm formation (Otto et al., 2001).

PheA, which is encoded by the gene XAC3467 and is secreted by both strains grown in XAM1 medium, is a chorismate mutase. PheA appears to be fundamentally important for plant–pathogen interactions because it catalyses the conversion of chorismate to prephenate, which is a precursor for the aromatic amino acids tyrosine (Tyr) and phenylalanine (Phe) (Kast et al., 2000). Chorismate is a substrate of the shikimate pathway, which regulates plant cell growth, development, structure, and pathogen defence. It can be converted into indole-acetic acid (IAA), salicylic acid (SA) and tryptophan (Bekal, Niblack & Lambert, 2003), culminating in the activation of plant defence systems (Weaver & Herrmann, 1997) (Fig. 3). Inactivation of the chorismate mutase gene in Ustilago maydis, the fungal agent that causes corn smut, drastically reduced symptoms associated with the disease (Djamei & Kahmann, 2012), demonstrating the importance of this protein to host–pathogen interactions. Finally, in bacteria of the genus Xanthomonas (Xanthomonas oryzae pv. Oryzae XKK.12), this protein has been reported to be directly involved in the process of virulence (Degrassi et al., 2010).

Figure 3 Model highlighting proteins related to virulence characterized in comparative proteomics.

Three of these proteins (A×21, FliC, and Ef-Tu) are characterized as PAMPs capable of inducing a PTI response, which can consequently lead to the increased synthesis of chorismate. Chorismate is a precursor in the synthesis of IAA, Trp, and SA, which are involved in the ROS response and induction of plant defence. However, the secretion of PheA may reduce plant defence responses by shifting the metabolism of chorismate to prephenate synthesis. Also highlighted is the Peh-1 protein, which is an endopolygalacturonase regulated by HrpX and cellulase (Egl), and endo-1,4-beta-glucanase (EngXca), which is regulated by RpfF. Endopolygalacturonase and cellulase degrade pectin polymers and cellulose in the plant cell wall, respectively. Phe, phenylalanine; Tyr, tyrosine; Trp, tryptophan; IAA, indole-3-acetic acid; SA, salicylic acid; PTI, (PAMP) -triggered immunity; PAMPs, pathogen-associated molecular pattern; ROS, reactive oxygen species; EF-Tu, elongation factor Tu, EFR - EF-Tu receiver; A×21, Ax21-triggered immunity; FliC, flagellin; FLS2, flagellin sensitive receiver 2; PheA, chorismate mutase; Peh-1, endopolygalacturonase; Egl and EngXca, cellulase. HrpG/HrpX and RpfC/RpfF, two component systems (sensory and regulatory proteins, respectively); Black, detected in Xac and ΔhrpB4 grown in XAM1; Red, detected exclusively in Xac grown in XAM1; Orange, detected in Xac grown in BOTH media (NB and XAM1); Blue, detected exclusively in Xac grown in NB. Dotted arrows indicate regulation.

Peptidyl-prolyl cis-trans isomerase (PPIase) is a protein encoded by the gene XAC1585. Similar to Slp, Peh-1, and PheA, this protein was identified in the extracellular matrix of both strains when grown in induction medium, confirming data described by Yamazaki, Hirata & Tsuyumu (2008). However, a paralogous protein was also secreted only by the mutant ΔhrpB4 grown in NB medium (Table S1). The PPIases play a pivotal role in catalysing the proper folding of many prokaryotic and eukaryotic proteins that are implicated in a variety of biological functions ranging from cell cycle regulation to bacterial infection (Pissavin & Hugouvieux-Cotte-Pattat, 1997; Zang et al., 2007). In Pseudomonas syringae and Xanthomonas, proteins orthologous to PPIase were identified in the secretome (Kaffarnik et al., 2009). In Xylella fastidiosa, which causes CVC, this protein is part of the composition of mature biofilms (Silva et al., 2011), and is differentially expressed when Xylella is subjected to thermal stress (Da Silva Neto et al., 2007).

The last protein with an annotated function secreted by both strains only when grown in XAM1 medium is a proteinase (XAC0552). The gene encoding this protein has been identified only in species that cause citrus canker; its genomic organization has been maintained in these species. In addition to containing a peptidase_S8 domain in its carboxyl-terminal region, it contains a Pro_kuma_activ domain (Pro-kumamolisin - PF09286.6) in its amino-terminal region, whose cleavage results in the activation of the remaining peptide (Comellas-Bigler et al., 2004). Although this domain has not been examined in the context of bacterial pathogen proteins, in the phytopathogenic fungi Mycosphaerella graminicola, expansion of the number of peptidases containing this domain may be associated with enhanced pathogenesis of wheat hosts (Goodwin et al., 2011).

Hypothetical proteins identified in this condition are encoded by the genes: XAC0223, XAC2562, XAC0677, and XAC0232. Of these, a protein worth noting is encoded by the XAC0223 gene (Fig. 3). This protein contains domains associated with the external membrane and a porin domain (Park & Ronald, 2012; Bahar et al., 2014). Orthologues of this gene in X. campestris pv. raphani 756C and X. oryzae pv. oryzicola have been reported to be activators of XA21-mediated immunity. Ax21 is secreted by a type I secretion system and in association with outer membrane vesicles (Bahar et al., 2014), and can act in a quorum-sensing system, motility, biofilm formation and virulence (Han et al., 2011). In X. oryzae pv. oryzicola, Ax21 was identified in the secretome, and deletion of the Ax21 gene resulted in reduced biofilm formation and extracellular polysaccharide production (Qian et al., 2013). Although classically the rice XA21 receptor directly recognizes Ax21 (Song et al., 1995), recent data have shown that other receptors, such as by Fls2, which is associated with a response in the presence of flagellin in Arabidopsis, can also recognize Ax21 secreted by Xanthomonas (Danna et al., 2011). This assumes that there is a dynamic between pathogens and microbe-associated molecular patterns in plants.

Although information about the functions of proteins encoded by the XAC2562, XAC0677, and XAC0232 genes is limited, domains associated with the function of lipoproteins were found in all of these proteins, which would justify their presence in the secretome.

Proteins secreted only by the wild-type strain grown in XAM1 medium

The data so far indicate that the production and secretion of important proteins associated with virulence and adaptation still occur in the ΔhrpB4 mutant. However, the impact of the mutation to the hrpB4 gene, which is associated with a loss of function of the T3SS system, is revealed by the identification of 14 unique proteins secreted only by wild-type Xac when cultured in XAM1 medium. Of these, nine had an annotated function (OprO, VirK, ribosomal proteins RplW, RplR and RpsE, HupB and cellulase Egl/Egl, EngXca), whereas the other five were hypothetical proteins (XAC0678, XAC3605, XAC1497, XAC3380 and XAC2682). Ten of the total amount of these proteins have a signal peptide sequence and only one hypothetical protein has a PIP-box sequence upstream of the start of the gene (Table 2). Although lacking annotated functions, all of these hypothetical proteins have a signal peptide sequence, suggesting that their secretion is critical for certain biological processes. Among the proteins with annotated functions, five are worth discussing further.

Table 2 Proteins expressed/detected only in wild type strain during infectious conditions.

					Strains (culture media)			
Gene ID	Gene name	Product	Cat.	NCBI ID	ΔhrpB4 (NB)	Xac (NB)	ΔhrpB4 (XAM1)	Xac (XAM1)	SP	PIP	
XAC3472	oprO	Polyphosphate-selective porin O	IV	1157543	−	−	−	+	Y	N	
XAC0678		Hypothetical protein	VIII	1154749	−	−	−	+	Y	Y	
XAC3605	uptE	Hypothetical protein	IV	1157676	−	−	−	+	Y	N	
XAC0435	virK	VirK protein	VII	1154506	−	−	−	+	Y	N	
XAC0974	rplW	50S ribosomal protein L23	III	1155045	−	−	−	+	N*	N	
XAC1497		Hypothetical protein	VIII	1155568	−	−	−	+	Y	N	
XAC0029	egl	Cellulase	VII	1154100	−	−	−	+	Y	N	
XAC1081	hupB	Histone-like protein	III	1155152	−	−	−	+	N*	N	
XAC0989	rpsE	30S ribosomal protein S5	III	1155060	−	−	−	+	N	N	
XAC3380		Hypothetical protein	VIII	1157451	−	−	−	+	Y	N	
XAC0028	egl	Cellulase	VII	1154100	−	−	−	+	Y	N	
XAC2682		Hypothetical protein	VIII	1156753	−	−	−	+	Y	N	
XAC0612	engXCA	Cellulase	VII	1154683	−	−	−	+	Y	N	
XAC0988	rplR	50S ribosomal protein L18	III	1155059	−	−	−	+	N	N	
Notes.

Gene_ID and Cat (primary category) according to Da Silva et al. (2002)

Product according to Kegg (Ogata et al., 1999)

SP Signal peptide

Y Yes

N No

N* Secreted by non-classical pathways

The OprO protein, encoded by the gene XAC3472, is an outer membrane porin specific for polyphosphates that can be regulated by the two-component regulatory systems, PhoPQ and PhoBR. In Pseudomonas aeruginosa, OprO is an anion-specific porin with a higher affinity for pyrophosphates than orthophosphates, and was expressed only under energy-poor growth conditions in phosphate during the stationary phase of bacterial growth (Siehnel, Egli & Hancock, 1992). Importantly, XAM1 medium lacks phosphate, similar to the mesophyll of citrus hosts. In recent work, Moreira and colleagues (2015). demonstrated that OprO, like other phosphate internalization systems, is expressed in Xac only during infection periods.

VirK, encoded by the gene XAC0435, is another protein secreted exclusively by the wild-type strain when cultured in XAM1 medium. This protein has a signal peptide for its secretion through the Sec pathway into the periplasmic space. In E. coli, VirK was characterized as a periplasmic protein required for the efficient secretion of plasmid-encoded toxins (Tapia-Pastrana et al., 2012). An orthologous gene found in Ralstonia solanacearum is regulated by HrpG. Microarray studies of the Xac transcriptome demonstrated that the expression of VirK was increased by more than two-fold when grown in XVM2 medium compared to when grown in non-inducing energy-rich culture medium (Astua-monge et al., 2005). The same study reported that a number of genes encoding secreted proteins were robustly expressed by XVM2 medium. Among the enzymes involved in cell wall degradation is cellulase (Egl), which is encoded by the genes XAC0029 and XAC0028. Both proteins were also secreted only by wild-type Xac in XAM1 medium; this was also true for the cellulase, EngXca. These proteins are predicted to be transported via the Sec pathway to the periplasm, and then secreted into the extracellular region by T2SS. Their secretion only by wild-type bacterium indicates that inactivation of T3SS in the mutant could suppress T2SS-secreted enzymes that degrade the cell wall of the host. Furthermore, EngXca has been specifically reported to be controlled by rpfF in Xanthomonas campestris. Poplawsky and colleagues (1998) and Siciliano and colleagues (2006) reported that Xac mutants, in which the rpfF and rpfC genes were inactivated, exhibited significantly decreased production of cell wall degrading enzymes and xanthan gum.

During the preparation of samples, the supernatant of the ΔhrpB4 strain cultured in XAM1 medium could be easily filtered through a 0.2 µm membrane, whereas the supernatant of the wild-type strain cultured in XAM1 medium could not be easily filtered. This is likely because the excess production of xanthan gum by wild-type bacteria obstructed the pores of the membrane. According to our studies, wild-type Xac is able to produce nearly 10 times more xanthan gum than the pathogenic deficient mutant ΔhrpB4 (mg gum/mg bacteria). Whereas wild-type Xac produces 46.21 ± 6.56A mg gum/mg bacteria, ΔhrpB4 produces only 5.21 ± 1.38B mg gum/mg bacteria (p value = 0.0004). These data underscore the importance of these genes to the formation of biofilm, as well as the importance of the hrpB4 gene in maintaining the activities of the T3SS and T2SS secretory systems.

The XAC0678 gene encodes a hypothetical protein but by sequence homology, it is predicted to encode a lipoprotein membrane protein. This hypothetical protein has a consensus PIP-box sequence upstream of the start of the gene, which indicates its probable regulation by HrpX. Furthermore, this protein is predicted to have a signal peptide for the Sec pathway, which could instruct its export into the periplasmic space.

Proteins secreted only by the mutant ΔhrpB4 grown in XAM1 medium

Ten proteins were secreted only by the mutant strain grown in XAM1 medium, 9 of which had annotated functions (Mlt, VacJ, OmpW, lei, GroES, YrbC, CycK, and PepN) and only one was predicted to be a hypothetical protein (XAC1761). Seven proteins had a signal peptide sequence, and a PIP-box sequence was not present upstream of any of these genes (Table 3).

Table 3 Proteins expressed/detected only in ΔhrpB4 strain during infectious conditions.

					Strains (culture media)			
Gene ID	Gene name	Product	Cat.	NCBI ID	ΔhrpB4 (NB)	Xac (NB)	ΔhrpB4 (XAM1)	Xac (XAM1)	SP	PIP	
XACb0007	mlt	Lytic murein transglycosylase	IV	1158494	−	−	+	−	Y	N	
XAC4344	vacJ	Lipoprotein	III	1158415	−	−	+	−	Y	N	
XAC3664	ompW	Outer membrane protein	IV	1157735	−	−	+	−	Y	N	
XAC1761		Hypothetical protein	VIII	1155832	−	−	+	−	Y	N	
XAC2781	leuS	Leucyl-tRNA synthetase	III	1156852	−	−	+	−	N	N	
XAC0541	groES	Co-chaperonin GroES	III	1154612	−	−	+	−	N*	N	
XAC1479		OmpA family protein	IV	1155550	−	−	+	−	Y	N	
XAC4342	yrbC	Toluene tolerance protein	VII	1158413	−	−	+	−	Y	N	
XAC2328	cycK	C-type cytochrome biogenesis membrane protein	I	1156399	−	−	+	−	Y	N	
XAC0645	pepN	Aminopeptidase	III	1154716	−	−	+	−	N*	N	
Notes.

Gene_ID and Cat (primary category) according to Da Silva et al. (2002)

Product according to Kegg (Ogata et al., 1999)

SP Signal peptide

Y Yes

N No

N* Secreted by non-classical pathways

Among these proteins, one worth noting is Mlt, which is a lytic murein transglycosylase. Xac has two copies of the mlt gene, XAC3225 and XACb0007, which are located on the chromosome and on the pXAC64 plasmid, respectively. The chromosomal copy of the gene is flanked by genes that encode the T3SS, XopE3 and XopAI effector proteins and the plasmid copy is flanked by the gene encoding XopE2 (Moreira et al., 2010). Mlt is highly similar to the hopAJ1 protein of Pseudomonas syringae, which was previously annotated as a T3SS helper protein (Oh et al., 2007). Mutations in the chromosomal copy reduced the ability of the mutant to cause canker (Laia et al., 2009).

The proteins encoded by the genes XAC3664 and XAC1479 are predicted to be outer membrane proteins. Although previously annotated as a hypothetical protein, the protein encoded by the gene, XAC1761, is predicted to be a lipoprotein based on its homology with sequences from other organisms.

Proteins secreted only by the wild-type strain grown in NB medium

Only 4 proteins were secreted by the wild-type strain grown in NB medium. Of these, 3 are directly associated with motility: PilY1, which is a Type IV pilus assembly protein, FliC (cover filament) and FliD (filament), which are structural proteins of the flagellum (Table 4). In addition, a hypothetical protein encoded by the gene XAC3314 was also secreted under these conditions. Of these four proteins, only PilY1 has a signal peptide sequence and a PIP-box sequence was not present upstream of any of these genes. PilY1 contains a DUF239 domain; these are found in a family of plant and bacterial proteins (PF03010). In addition, orthologous proteins were only found in Xac29, XacW, and Xcv, in Xoo MAFF and Xoo PXO, a protein similar to transposase. Mutant Pseudomonas strains, in which the pilY1 gene was inactivated, exhibited robust suppression of the swarming motility (Kuchma et al., 2010). In Xylella fastidiosa, a mutation in the pilY1 gene led to a reduction, but not a complete loss, of type IV pilus apparatus assembly and twitching motility (Li et al., 2007). In Xoo, pilY1, in combination with other genes involved in adhesion and biofilm formation, was upregulated six days after infection in the plant (Soto-Suárez et al., 2010).

Table 4 Proteins expressed/detected only in wild type strain during non-infectious conditions.

					Strains (culture media)			
Gene ID	Gene name	Product	Cat.	NCBI ID	ΔhrpB4 (NB)	Xac (NB)	ΔhrpB4 (XAM1)	Xac (XAM1)	SP	PIP	
XAC2665	pilY1	Type IV pilus assembly protein PilY1	IV	1156736	−	+	−	−	Y	N	
XAC1975	fliC	Flagellin	V	1156046	−	+	−	−	N*	N	
XAC3314		Hypothetical protein	VIII	1157385	−	+	−	−	N*	N	
XAC1974	fliD	Flagellar hook-associated protein 2	V	1156045	−	+	−	−	N*	N	
Notes.

Gene_ID and Cat (primary category) according to Da Silva et al. (2002)

Product according to Kegg (Ogata et al., 1999)

SP Signal peptide

Y Yes

N No

N* Secreted by non-classical pathways

FliC and FliD are protein subunits that form the structure of flagella. FliC, also known as flagellin, forms the flagellar filament itself, and is therefore abundantly synthesized by flagellated bacteria. FliC is directly associated with the innate immune response of plants, since it has the ability to bind to specific receptors, such as Toll-like receptor 5 (TLR5) (Hayashi et al., 2001). More recently, part of the N-terminal portion of this protein, corresponding to 22 amino acids (flg22) and conserved across a wide range of pathogens, was shown to activate plant defence mechanisms (Navarro et al., 2004). This 22-amino acid protein fragment is recognized by specific receptors such as by the kinase FLS2, which in turn activate the MAPK signalling cascade (mitogen-activated protein kinase), which is capable of regulating the expression of over 900 genes in the Arabidopsis thaliana genome (Asai et al., 2002).

Protein secreted only by the wild-type strain in both growth media

The only protein detected exclusively in the secretome of the wild-type strain when grown in either growth media was the EF-Tu protein encoded by the gene XAC0957 (Table S1).

EF-Tu is also characterized as a PAMP, like FliC (Nürnberger & Kemmerling, 2006). Although EF-Tu has a fundamental role in protein synthesis, a diversity of other functions have been attributed to this protein. In Burkholderia pseudomallei, EF-Tu was characterized as a membrane-associated protein that is secreted by the formation of outer membrane vesicles (OMV), and is therefore an important target in the development of vaccines against this pathogen (Nieves et al., 2010). In E. coli, EF-Tu acts as a chaperone molecule by binding to denatured proteins after stress conditions (Caldas, El Yaagoubi & Richarme, 1998). In Agrobacterium tumefaciens, EF-Tu can act as an inducer in Arabidopsis, whose specific receptor is the EFR protein (Zipfel et al., 2006). EF-Tu has also been reported to be critical for biofilm formation in Xylella (Silva et al., 2011) and, more recently, in Xac (Zimaro et al., 2013). Considering all of these functions attributed to this protein and the inability of the ΔhrpB4 mutant to secrete this protein under all conditions tested, we hypothesize that the secretion of this protein may be associated with T3SS in Xanthomonas. Besides its participation in the production of biofilms, EF-Tu may have another key role in the virulence process, and may represent a potential moonlighting protein (Jeffery, 1999; Henderson & Martin, 2011).

All other secreted proteins detected in our study

In addition to the proteins described above, 13 other secreted proteins were detected in this study (Table S1). Because these proteins were detected in specific but restricted conditions, we could not exclusively associate them with a specific growth media or specific bacterial strain. The majority (8) of these have enzymatic functions (amylase, proteases, and isomerase). Two membrane-associated proteins (OMPs), and two hypothetical proteins, XAC0753 and XAC0868, were also identified. The latter two proteins have domains with peptidase activity, although XAC0868 has a more representative ricin B lectin domain. Interestingly, the protein encoded by XAC0868 was the only protein detected in all four conditions tested and exists only in Xanthomonas strains that infect citrus plants. A model highlighting proteins related to Xac adaptation was devised (Fig. 4).

Figure 4 Model highlighting proteins related to adaptation in comparative proteomics.

These proteins were grouped into five groups: lipoproteins, porins, outer membrane proteins, secreted proteins, and ribosomal proteins. HrpG/HrpX, PhoQ/PhoP and PhoR/PhoB—two component systems (sensory and regulatory proteins, respectively); Black, detected in Xac and ΔhrpB4 grown in XAM1; Red, detected exclusively in Xac grown in XAM1; Orange, detected in Xac grown in both media (NB and XAM1); Blue, detected exclusively in Xac grown in NB. Dotted arrows indicate regulation; Pi/PolyPi, inorganic phosphate and inorganic polyphosphate.

Pathogenicity of Xac compared to other plant pathogens

To better understand the importance of secreted proteins in the virulence of plant-specific pathogens of the Xanthomonadaceae family, a comparative analysis of proteins identified in this study with proteins identified in six other proteomic studies was performed (Table 5). Despite the diversity of model organisms studied and methodologies used, orthologous proteins were identified in these studies.

Table 5 Xac annotated proteins expressed and secreted compared to other studies of Xanthomonadaceae family organisms.

Function/product	Mesophyllic	Vascular	
	Xacb	Xacc	Xacd	Xooe	Xoof	Xccg	Xfh	
Hydrolitic enzymes	
Alpha-amylases	X							
Aminopeptidases	X							
Cellulases	X					X		
Endopolygalacturonases	X	X						
Extracellular proteases	X							
Proteinases	X	X						
Cysteine protease	X	X			X			
Serine Proteases	X							
Motility/chemotaxis	
Flagellar associated protein	X							
Flagellina	X			X	X	X		
Type IV pilus assembly protein PilY1a	X	X		X				
Twiching motility					X		X	
Adaptation/Virulence	
Lytic murein transglycosylasea	X							
RpfNa	X		X					
T3SSe				X		X		
ROS				X	X			
Elongation factor Tua	X		X	X	X	X	X	
Membrane associated proteins	
TBDRs			X	X		X		
Iron/siderophores receptors				X	X	X		
ABC transporters				X				
Lipoproteins	X							
Outer membrane proteins	X		X	X	X	X	X	
Polyphosphate-selective porin Oa	X				X			
Toluene tolerance proteins	X							
Others	
Chorismate mutasea	X							
Co-chaperonin GroESa	X		X	X				
C-type cytochrome biogenesis membrane proteina	X							
Histone-like proteina	X							
Leucyl-tRNA synthetasea	X							
VirK proteina	X	X		X	X			
Polyvinyl alcohol dehydrogenasea	X							
Molecular chaperone GroELa	X		X	X	X		X	
Peptidyl-prolyl cis-trans isomerasea	X			X	X		X	
Ribosomal proteins	X		X	X	X			
ATP synthase			X		X		X	
Notes.

a Unique proteins.

b This work (reference data).

c Yamazaki, Hirata & Tsuyumu (2008).

d Zimaro et al. (2013).

e Wang et al. (2013).

f González et al. (2012).

g Sidhu et al. (2008).

h Silva et al. (2011).

Yamazaki, Hirata & Tsuyumu (2008) studied the role of the HrpG gene in the regulation of the synthesis of Xac proteins secreted by T2SS in energy-rich culture medium. In these studies, a total of 11 proteins were identified, six of which were extracellular enzymes. Zimaro and colleagues compared the proteome of planktonic Xac cells and Xac biofilm cells, and identified 53 differentially expressed proteins, of which 31 were upregulated in biofilms. Silva and colleagues (2011) identified 456 proteins potentially related to biofilm formation in Xylella fastidiosa. Wang and colleagues analysed the Xanthomonas oryzae pv. oryzae secretome grown in culture medium and in plants, and identified 109 unique proteins. Proteins involved in pathogenicity and the induction of host defence were identified only in the secretome of bacteria in contact with the plant (Yamazaki, Hirata & Tsuyumu, 2008). Using the same model study, Gonzalez and colleagues identified 324 different proteins secreted by Xoo in contact with the xylem of the host plant. Of these, 64 were suggested to be associated with the process of virulence (González et al., 2012). Sidhu and colleagues (2008) identified 31 proteins associated with OMV, half of which were associated with the induction of virulence in Xanthomonas campestris pv. campestris.

This comparative analysis revealed the correlation of a number of proteins with the life cycle or the infection model of the associated pathogen. For instance, endopolygalacturonase and proteinases, which seem to be crucial to the virulence process, are induced by mesophilic strains. Proteins associated with motility and chemotaxis have been identified in virtually every proteomics study included in our analysis, which is consistent with the importance of these proteins in the formation of cellular aggregates, biofilms or even for inducing a response in the host plant (Malamud et al., 2011; Dunger et al., 2014). Mlt was detected only in this study, indicating its importance to the induction of virulence in Xac. Interestingly, EF-Tu was independently identified in five other studies, including in the Xf strain, whereas GroEL was detected in four other studies, and flagellin (FliC), VirK and peptidyl-prolyl cis-trans isomerase (PPI) were identified in three other studies. It is important to note that FliC is absent in the Xylella genome since it does not have flagellum. The possibility that GroEL, FliC, Ef-Tu and PPI can act as moonlighting proteins in bacteria of the family Xanthomonadaceae is highlighted (Henderson & Martin, 2011).

Conclusions

Our results indicate that the secretome analysis is a valuable approach to highlight proteins potentially involved with the virulence in Xanthomonas. Fifty-five unique proteins were identified in total on the secretomes of wild-type Xac and the mutant ΔhrpB cultured in different growth media. Of these, 37 have a predicted signal peptide for secretion via the general secretory pathway and are potentially related to the pathogenicity and adaptation during infections and plant–pathogen interactions. However, of the 18 proteins lacking a signal peptide, 13 were predicted to be secreted by non-classical routes. This demonstrates that the majority (50) of proteins detected in this study were neither contaminant proteins nor proteins released by cell lysis. Interestingly, the percentage of secreted proteins related to the pathogenicity, virulence and adaptation increased nearly two-fold (from 15 to 29%) when considering only proteins secreted by wild-type Xac bacterium compared to those secreted by its mutant ΔhrpB4. These results corroborate that the hrpB4 gene act as T3SS central player, impacting the emergence of pathogenicity in Xanthomonas. For instance, the expression/secretion of several T2SS proteins appears to be regulated by T3SS activity. Our results indicate that this regulation also occurs in Xac. The profile of proteins secreted exclusively by wild-type Xac and the increased amount of xanthan gum produced in XAM1 medium suggests that DSFs (diffusible signalling factors) responsible for controlling these factors may be related to the lack of expression of the HrpB4 protein, since the ΔhrpB4 mutant, which is deficient in the production of this protein, was unable to secrete several of these factors. Further analysis are necessary to confirm that hypothesis.

Overall, the secretome method allows the testing of not only one, but several mutants at a time. This may be a powerful tool to shed light into the biology of Xanthomonas. Indeed, analyses are underway with different mutants generated under different mutagenic methodologies to evaluate others gene targets that might be related to the disease.

Supplemental Information

Table S1 Supplemental table 1

Click here for additional data file.

We thank all the members of the Laboratory of Molecular Biology of the Department of Technology (UNESP—Universidade Estadual Paulista—Jaboticabal). We would also like to thank the Brazilian Synchrotron Light Laboratory.

Additional Information and Declarations

Competing Interests

Author Contributions

Data Availability

The authors declare there are no competing interests.

Rafael M. Ferreira and Leandro M. Moreira conceived and designed the experiments, performed the experiments, analyzed the data, wrote the paper, prepared figures and/or tables, reviewed drafts of the paper.

Jesus A. Ferro conceived and designed the experiments, analyzed the data, contributed reagents/materials/analysis tools, wrote the paper, reviewed drafts of the paper.

Marcia R.R. Soares and Julio C.F. de Oliveira conceived and designed the experiments, performed the experiments, analyzed the data, reviewed drafts of the paper.

Marcelo L. Laia conceived and designed the experiments, analyzed the data, reviewed drafts of the paper.

Alessandro M. Varani analyzed the data, contributed reagents/materials/analysis tools, wrote the paper, reviewed drafts of the paper.

Maria Ines T. Ferro analyzed the data, contributed reagents/materials/analysis tools, reviewed drafts of the paper.

The following information was supplied regarding data availability:

Zenodo: https://zenodo.org/record/45560#.VrNQg4RSzCc.

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
