# Peer review of "Unravelling potential virulence factor candidates in Xanthomonas citri. subsp. citri by secretome analysis"

_PeerJ, doi:10.7717/peerj.1734_

## Round 0.1 · original submission · Minor Revisions

· Academic Editor

Minor Revisions

This work provides a method and approach for building secretome analysis of Xanthomonas. As mentioned in the manuscript some of the issues associated with the xanthan gum may be somewhat troublesome in collecting easily compared samples. Though your minimal XAM1 medium has a reputation for mimicking a close approximation of a testable system between the wild-type and mutant, the nutrient rich NB medium may be too different to contrast the two focused on for this study; however, it did present you with different candidates to focus your attention on. Perhaps in later experimentation you might tweak the nutrients present in a minimal medium, and surely additional mutants would be beneficial in contrasting candidates believed important in the interaction; to some degree you attempted extracting roles of candidates identified in other systems. In general, the manuscript is well written, but some attention to having a discussion and conclusion section might be sorted out better for clarity. Please note some of the other reviewers suggestions; I am forwarding this as requiring “minor revision”, but it would be valuable to re-organize and perhaps revise the presentation style as suggested. Considering the difficulties associated with this genus, you provide valuable insight in your approach and its use in attempting to catalog gene function. Congratulations on your efforts. Highlighted below are a few additional notes:

Example of annotation:
LINE NO.: / PREVIOUS FORM / SUGGESTED FORM / [ADDITIONAL NOTES]
or
LINE NO. note: Additional suggestions.

36 note: initial sentence; strain 306 is one of the strains, not the only causal strain. There are other strains which affect citrus production, yes?

55: / secretome / secretome data /[]

56: / which in ultimate / which ultimately /[]

91: / including for bacteria / including bacteria /[]

95: / pest / microorganism /[classified as pest under agency wide interpretation, but might be re-worded]

216: / MASCOT //[Are annotations for sequences based on this program, UNIPROT, or other?]

244: /medium//[Is there a designated name for this medium other than the recipe?]

273: / annotated / MASCOT annotated /[what is the source of the annotation? see 216:]

409 note: Why so few mutants available or used? Does this represent the testing capability is using the secretome analysis methods? Can other mutants, or are there plans to track town influence on other mutants on pathogenicity/secretion? Maybe some sort of plan to move forward can be placed in the discussion.

473 note: Is plasmid content considerably different in different pathogen isolates, and does the plasmid still exist in the Tn5 mutant? Can difference be explained by plasmid presence as has been associated in other microbial cases.

576 note: You go directly into Conclusions; as there is no hard page limit it might be a good opportunity to include a discussion section to highlight your observations which were presented in your different Results observations.

960 note: I see symbols for unique proteins and reference data (a), but I don't see explanations for the superscripts b-g highlighted in the columns.

Reviewer 1 ·

Basic reporting

General view:
The ms “Unravelling potential virulence factor candidates in Xanthomonas citri. subsp. citri by secretome analysis” by Ferreira et al. is a very descriptive work comparing two genetic background of Xanthomonas citri, the wild type strain and the hrpB4 mutant generated by the group. It is a valuable work that I believe is intended to guide future functional analyses with this plant pathogen.
The ms is in general well written, but the way in which the data was presented needs attention
For such a descriptive kind of analysis, sometimes it is difficult to separate results and discussion. I suggest to blend the two sections, results and discussion, in one.

Style:
Check the nomenclature of Tn5::kan and be consistent throughout the text

English needs minor revision

Figure 2 legend needs to be re-written for clarity. The way it is presented it is difficult to understand at first look what the circles mean- maybe you could say A1 and A2. Are the two circles on the left representing total proteins detected?
In the text and legend authors use CN and NB- choose one format.

Results/discussion:
1. Do the authors know what is coded by hrpB4? Is there a function assigned to HrpB4 (recently)?
2. From line 259: it is interesting that exactly 55 proteins were found in both secretomes. Or perhaps 55 proteins were found in total? I think the authors wanted to start with statistics of total proteins detected, and if so, re-write for clarity. Overall, authors were extremely concise in results and it is somewhat confusing the number of proteins detected in a specific situation.
3. Line 278- authors mix results and discussion- from the beginning it looks results; perhaps for this kind of descriptive data authors could fuse results and discussion in one section
4. Still from line 278- 13 proteins that are common to both? Or 13 proteins secreted by both in total? Make it clear for the reader that is not used to your work.
5. I could not find where figure 4 is called in the text

Experimental design

Experimental design is appropriate
Methodology is well defined and clear

Validity of the findings

As I mentioned above, the work is sound, and despite being very descriptive it may guide future work that could lead to the understanding of the function of HrpB4
Perhaps the authors could rise some possible uses for these findings

Reviewer 2 ·

Basic reporting

The manuscript entitled “Unravelling potential virulence factor candidates in Xanthomonas citri. subsp. citri by secretome analysis” authored by R.M. Ferreira et al. presents a complete introduction and background related with the citrus canker focused in the causal agent of this disease Xanthomonas citri subspp. citri. In this work the secretomes of two different strains from Xanthomonas citri (Xac wild type and a mutant hrpB4) grown in different media are presented and compared. A complete picture of the bacterial secretory systems is presented demonstrating that the use of secretome is a valuable approach that will bring more knowledge of the biology of bacterial plant pathogen.
On the other hand a full survey of the transcriptomes of the Xanthomonadaceae family is done providing a complete picture of the different global expressions of bacterial proteins from Xanthomonas grown in different conditions. Furthermore this approach could allow the design of new strategies to combat plant pathogens.
Figures presented are appropriate to understand the results obtained, but a better resolution of some of them would be desirable. In Figure 1, for example, the picture of infected citrus leaf could be replaced with a better resolution.

Experimental design

The paper is properly presented and provides a comprehensive and a deeper analysis of the secretomes of Xanthomonas grown in different conditions .
The study is well structured, based on a thorough methodological approach and results are adequately presented and interpreted.
However, there are some issues that if revised or corrected may improve the work:
1) A significant reduction in the discussion is necessary, to make it easy and understandable the comprehension of the key and important results obtained.
2) How the proteomics assays/experiments were performed proteomic to ensure reproducibility of the results.
3) Data from xanthan production need a statistical analysis in order to ensure that the different amount of xanthan obtained in the wild type and mutant strains from Xac are statistically significant.

Validity of the findings

The conclusions are clearly connected with subject under investigation. But the written presentation of the conclusion needs to be modified in a way that the most relevant results are easier to read and understand.
The results obtained represented an advance in the study of the molecular basis of plant-pathogen interaction, specifically in the citrus canker disease.

Additional comments

For the mentioned reasons I consider that this paper is suitable for publication in PeeJ after a minor revision.

---

## Round 0.2 · accepted · Accept

· Academic Editor

Accept

Your latest revision reads very well and the organization of the topics appears to flow much better and I now recommend this version as ready for publication. Reviewer recommendations were sufficiently accommodated. The description of your methodology for secretome analysis may set the stage for thorough screening of mutants and identification of key candidates involved in plant-microbe interaction studies. With in planta screening being the desired test environment, this simulated system may assist in wide-scale screening before keying in on the critical candidates for detailed analysis. Thank you for the manuscript submission.